# Methodological establishment and diagnostic value of a multiplex fluorescent PCR assay for the detection of three fastidious respiratory pathogens

Jingchao Shi[1], Yijun Zhu [1]*, Shuyun Chen[2,3], Xiaoyun Shan[1], Lihong Bo[1], Kai Shen[1], Keqiang Chen[1]

**1** Department of Clinical Laboratory, Affiliated Jinhua Hospital, Zhejiang University School of Medicine (Jinhua Municipal Central Hospital), Jinhua, Zhejiang Province, China, **2** Hangzhou Dian Medical Center, Hangzhou, Zhejiang Province, China, **3** Key Laboratory of Digital Technology in Medical Diagnostics of Zhejiang Province, Hangzhou, Zhejiang Province, China

* zhuyijunwz@sina.com

## Abstract

### Background

*Streptococcus pneumoniae*, *Haemophilus influenzae*, and *Moraxella catarrhalis* are common fastidious bacteria responsible for respiratory tract infections, particularly in children and immunocompromised individuals. Due to their demanding growth requirements, traditional culture methods often yield low sensitivity and delayed results, posing challenges for early and accurate diagnosis.

### Objective

To establish a TaqMan probe-based multiplex fluorescent PCR method for the simultaneous rapid detection and identification of three important respiratory fastidious pathogens: *Streptococcus pneumoniae*, *Haemophilus influenzae*, and *Moraxella catarrhalis*.

### Methods

By designing and optimizing TaqMan probes and primers targeting *Streptococcus pneumoniae*, *Haemophilus influenzae*, and *Moraxella catarrhalis*, the sensitivity, specificity, and reproducibility of the TaqMan probe-based multiplex fluorescent PCR assay were examined. A total of 173 clinical samples which are sputum and alveolar lavage fluid were tested simultaneously using traditional culture methods and multiplex fluorescent PCR assays, and the results were compared for consistency. For samples with inconsistent detection results, targeted high-throughput sequencing (tNGS) was used for confirmation.

**Data availability statement:** The primer and probe sequences used in this study have been deposited in the BioStudies repository, with the accession number S-BSST1515 and can be accessed via https://www.ebi.ac.uk/biostudies/studies/S-BSST1515.

**Funding:** This work was supported by the Science Technology Department of Zhejiang province, China (LGC22H200018), and the Jinhua Science and Technology Bureau, Zhejiang province, China (2021-3-026). The funders participated in the study design, data collection, manuscript writing, and decision to publish.

**Competing interests:** The authors have declared that no competing interests exist.

**Abbreviations:** SP, *Streptococcus pneumonia*; HI, *Haemophilus influenza*; MC, *Moraxella catarrhalis*; PCR, Polymerase Chain Reaction; tNGS, Targeted Next-Generation Sequencing; ATCC, American Type Culture Collection; Ct, Cycle threshold; MGB, Minor Groove Binder; DNA, Deoxyribonucleic acid; rRNA, ribosomal Ribonucleic Acid; SYBR, SYBR Green (a nucleic acid dye used in molecular biology).

## Results

The limit of detection of the TaqMan probe-based multiplex fluorescent PCR method was 100 Copies/ml for *Streptococcus pneumoniae*, 20 Copies/ml for *Haemophilus influenzae*, and 50 Copies/ml for *Moraxella catarrhalis*. The detection rate of the multiplex fluorescent PCR method was in high concordance with traditional culture methods and tNGS in the 173 clinical samples (Kappa = 0.819). The multiplex fluorescent PCR method demonstrated high sensitivity and specificity, detecting more cases of mixed infections.

## Conclusion

The multiplex fluorescent PCR method established in this study provides a powerful tool for rapid and accurate clinical detection of fastidious respiratory pathogens, with significant clinical application value and promotion prospects. This method offers substantial advantages over traditional culture methods in terms of detection speed and sensitivity, particularly in detecting mixed infections, and has significant potential for clinical application and widespread adoption.

## Introduction

*Streptococcus pneumoniae*, *Haemophilus influenzae*, and *Moraxella catarrhalis* are common fastidious bacteria that cause respiratory tract infections in the community. Fastidious bacteria are a large group of bacteria with demanding growth and nutritional requirements, making them difficult to culture in ordinary environments. In vitro cultures require special factors or additional nutrients. *Streptococcus pneumoniae* is a potentially lethal human pathogen associated with high morbidity, mortality, and a significant global economic burden [1]. *Haemophilus influenzae* is a major pathogen causing respiratory infections and invasive diseases, resulting in considerable disease burden [2]. *Moraxella catarrhalis* survives only on the mucosal surfaces of the respiratory tract in humans and does not usually cause infections, but can lead to respiratory infections under certain conditions [3]. The human upper respiratory tract, extending from the nostrils to the portion of the larynx above the vocal cords, has the highest bacterial density in the entire respiratory system. *Streptococcus pneumoniae*, *Haemophilus influenzae*, and *Moraxella catarrhalis* are also the most common causative organisms of acute otitis media, acute sinusitis, and persistent bacterial bronchitis in children [4–6].

Mixed infections involving two or more of these pathogens are increasingly recognized in clinical practice, especially in children, the elderly, or patients with underlying viral infections and immunosuppression. Such polymicrobial infections may lead to more severe respiratory symptoms, prolonged disease duration, and poorer prognosis compared to single infections. They also complicate antimicrobial treatment strategies due to overlapping resistance patterns and possible synergistic interactions between pathogens [7,8].

These pathogens have distinct virulence mechanisms: *S. pneumoniae* produces a polysaccharide capsule that inhibits phagocytosis and aids immune evasion; *H. influenzae* secretes IgA proteases and invades epithelial cells; and *M. catarrhalis* is capable of biofilm formation and produces β-lactamases to resist antibiotics [9–11]. The host immune response includes neutrophil recruitment, mucosal antibody secretion, and pro-inflammatory cytokine production. However, in mixed infections, this response can be dysregulated, potentially exacerbating tissue damage or delaying resolution [12].

In many cases, empirical antibiotic therapy is initiated before the causative pathogen is identified, especially in community-acquired lower respiratory tract infections, due to the need for timely treatment and limitations in rapid diagnostics [13]. This practice has led to the overuse of antibiotics, contributing to the emergence of β-lactamase-producing strains of *Haemophilus influenzae* and *Moraxella catarrhalis*. The current reference method for isolating and characterizing these pathogens is traditional laboratory culture methods, especially in primary care. However, these bacteria are fastidious, and bacterial culture methods often lack sensitivity, are time-consuming, labor-intensive, and can take 24–48 hours to yield results, thus lengthening the testing cycle and reducing clinical utility. Polymerase chain reaction (PCR)-based methods have been shown to be rapid, low-cost, and highly sensitive, often applied directly to clinical samples from sterile and non-sterile sites to detect pathogenic bacteria [14]. However, detecting these pathogenic bacteria usually requires multiple single PCR tests per patient, which is both expensive and inefficient. Multiplex real-time PCR methods allow rapid and simultaneous detection of multiple respiratory pathogens in clinical specimens. The two main types of multiplex real-time PCR currently available are fluorescent probe-based assays and fluorescent dye-based assays. Since SYBR Green binds to any double-stranded DNA, it can produce false-positive signals in the presence of non-specific amplification products or primer dimers, thus requiring optimization of PCR conditions. Therefore, TaqMan probes, which are more specific and suitable for multiplex amplification, have an advantage over the SYBR Green dye method [15].

The aim of this study was to establish a TaqMan probe-basedmultiplex fluorescent PCR assay for the three aforementioned pathogenic bacteria, providing a reference for the rapid and accurate clinical detection of mono or mixed infections caused by *Streptococcus pneumoniae*, *Haemophilus influenzae*, and *Moraxella catarrhalis*.

## Materials and methods

### Test materials

**Experimental strains.** *Streptococcus pneumoniae* (ATCC 49619), *Haemophilus influenzae* (ATCC 49247), and *Moraxella catarrhalis* (ATCC 25238) were purchased from Wenzhou Kangtai Biotechnology Co. *Escherichia coli* (ATCC 25922), *Pseudomonas aeruginosa* (ATCC 27853), and *Staphylococcus aureus* (ATCC 25923) were purchased from the National Institute for the Control of Pharmaceutical and Biological Products. Clinical strains were isolated from sputum or alveolar lavage specimens of patients from April 2022 to August 2023 at Jinhua Central Hospital. All pathogenic microorganisms were stored at −80°C.

**Experimental setup.** For our experiments, we utilized a range of essential reagents and instruments. Detailed information on these is provided in the supplementary materials, specifically in S1 Table.

All reagents and plastic consumables used were certified DNase/RNase-free and handled in a designated clean area to minimize the risk of cross-contamination. Negative controls were included at each extraction and amplification batch. PCR setup was performed in a biosafety cabinet using filtered tips and separated unidirectional workflows.

### Test methods

**Design of multiplex real-time RT-PCR primer-probe pairs.** The genome sequences of *Moraxella catarrhalis*, *Haemophilus influenzae*, and *Streptococcus pneumoniae* were downloaded from public databases. To ensure the specificity and sensitivity of the newly developed assays, we conducted a comprehensive sequence comparison using the Vector NTI software. The genome sequences of *Moraxella catarrhalis*, *Haemophilus influenzae*, and *Streptococcus pneumoniae* were analyzed to identify unique and conserved regions suitable for primer and probe design. For *Moraxella*

*catarrhalis*, the copB gene was selected as the target due to its specific presence and variability among strains. For *Haemophilus influenzae*, the fucK gene was chosen for its stability and uniqueness in the species. Similarly, for *Streptococcus pneumoniae*, the lytA gene was identified as an optimal target due to its high specificity and frequent use in diagnostic assays. The designed primers and probes based on these targets are listed in Table 1, and their performance was validated through a series of experiments to ensure high specificity and amplification efficiency. The primers and probes were synthesized by Briggs Biotechnology (Shanghai) Co. and diluted to 10 µmol/L before use.

Each primer and probe batch underwent analytical validation for specificity and efficiency prior to use in multiplex assays.

### Extraction of bacterial DNA

**Extraction of DNA from standard strains:** Bacteria were inoculated in blood plate medium, cultured overnight at 37°C with 5% CO2. Colonies were picked and suspended in 1 ml of saline, centrifuged at 12,000 rpm for 2 minutes, and the supernatant was discarded. Bacterial genomic DNA was extracted using the Tengen Bacterial Genomic DNA Extraction Kit (centrifugal column type), following the kit instructions.

**Extraction of DNA from clinical specimens:** Equal amounts of sputum liquefaction solution (Oxoid, UK) were added to sputum or alveolar lavage fluid, mixed well by shaking, then 1 ml of the mixture was taken into a centrifuge tube. Samples were centrifuged at 12,000 rpm for 2 minutes, and the supernatant was discarded. Bacterial genomic DNA was extracted using the Tengen Bacterial Genomic DNA Extraction Kit (centrifugal column type), following the kit instructions.

Quality Control Measures:

1 Each batch of DNA extraction included a negative extraction control (sterile water) to monitor for reagent contamination.

2 DNA quantity and purity were measured using a Nanodrop spectrophotometer (A260/280 ratio) to ensure sample integrity.

3 All extracted DNA was aliquoted and stored at −20 °C to prevent repeated freeze–thaw cycles.

### Establishment of multiplex fluorescent PCR assay

A 40 µl reaction system was used: 20 µl of 2×Accurate TaqHS Probe Premix (UNG Plus), 0.4 µl each of upstream primer, downstream primer, and probe (all at 10 µM concentration) for each target, 5 µl of template, and 11.4 µl of ddH$_2$O. Optimized conditions (annealing time/temperature, primer, and probe concentrations) for primer-probe pairs against *Streptococcus pneumoniae*, *Haemophilus influenzae*, and *Moraxella catarrhalis* were verified using single-plex PCR reactions. Experiments were performed using the SLAN fully automated medical PCR analysis system. Samples were preheated at 95°C for 3 minutes for complete DNA denaturation, followed by 42 cycles of 95°C for 15 seconds for DNA denaturation, and 60°C for 45 seconds for primer-DNA binding and extension. Fluorescence signals were recorded during the annealing phase of the 42 cycles. Finally, a post-extension step was performed at 37°C for 10 seconds to ensure the stability

**Table 1. Primer-probe information for *Streptococcus pneumoniae*, *Haemophilus influenzae*, and *Moraxella catarrhalis*.**

| Species | Target Gene | Primer/Probe sequence (5′ to 3′) | Size (bp) | 5′ end labeling | 3′ end marking | Reference |
|---|---|---|---|---|---|---|
| *Streptococcus pneumoniae* | lytA | F: GCACGAATAACCAACCAAACA<br>R: AATCGTCAAGCCGTTCTCAAT<br>P: GGCATTAGCCGTGAGCAGTT | 116 | Cy5 | MGB | This study |
| *Haemophilus influenzae* | fucK | F: CCATTTCCCTCCTATGCGTTAT<br>R: TGATTCAGCCCTGCACCAG<br>P: CGTACACCGTTAGCCC | 150 | HEX | MGB | This study |
| *Moraxella catarrhalis* | cpoB | F: TGCCAAAGACACCTCCATCAT<br>R: GTTGCCATAAGGCTTCCAGTT<br>P: GTGAAGGCTATAATGT | 115 | FAM | MGB | This study |

of the amplified products. Ct values were determined by the SLAN system under automatic threshold settings, with Ct values ≤ 40 defining positivity. Each multiplex PCR run included a negative and a positive control. The positive control consisted of a mixture of nucleic acids of known concentration extracted from *Streptococcus pneumoniae*, *Haemophilus influenzae*, and *Moraxella catarrhalis* strains.

### Sensitivity testing

DNA from *Streptococcus pneumoniae*, *Haemophilus influenzae*, and *Moraxella catarrhalis* was amplified by multiplex real-time RT-PCR starting from $1.5 \times 10^6$ Copies/ml in the following decreasing concentration order: $1.5 \times 10^5$, $1.5 \times 10^4$, $1.5 \times 10^3$, 500, 200, 100, 50, 20, 10 Copies/ml. The lowest detectable concentration was obtained based on the resulting amplification curve.

### Specificity testing

Using the DNA of *Streptococcus pneumoniae* (ATCC 49619), *Haemophilus influenzae* (ATCC 49247), and *Moraxella catarrhalis* (ATCC 25238) as positive controls and ddH$_2$O as a negative control, the newly established assay was applied to analyze the DNA of standard strains such as *Escherichia coli* (ATCC 25922), *Pseudomonas aeruginosa* (ATCC 27853), *Staphylococcus aureus* (ATCC 25923), and other standard strains, as well as *Streptococcus pseudopneumoniae*, *Streptococcus mitis*, *Haemophilus parainfluenzae*, *Haemophilus haemolyticus*, *Haemophilus parahaemolyticus*, *Moraxella osloensis*, *Streptococcus salivarius*, *Streptococcus oralis*, *Corynebacterium striatum*, *Stenotrophomonas maltophilia*, *Staphylococcus haemolyticus*, *Clostridium perfringen*, *Acinetobacter baumannii*, *Enterococcus faecalis*, *Klebsiella pneumoniae*, *Acinetobacter pittii*, *Enterococcus avium*, *Klebsiella oxytoca*, *Neisseria meningitidis*, *Legionella pneumophila*, *Bordetella pertussis*, *Staphylococcus epidermidis*, and *Bordetella bronchiseptica* to evaluate the specificity of the method.

### Repeatability test

The DNA of *Streptococcus pneumoniae*, *Haemophilus influenzae*, and *Moraxella catarrhalis* was subjected to 10-fold serial dilutions based on the standard concentrations used in the sensitivity testing. Three representative concentrations were selected for each species, corresponding to high, moderate, and low template levels. These levels produced Ct values approximately in the ranges of 24–25, 28–29, and 31–35, which are estimated to correspond to $\sim 10^6$, $\sim 10^3$–$10^4$, and $\sim 10$–100 copies/mL, respectively. For each concentration, three replicate multiplex fluorescent PCR reactions were performed. The mean, standard deviation, and coefficient of variation (CV) of the Ct values were calculated to assess assay repeatability. Acceptance criteria for repeatability required CV < 3%.

### Clinical sample testing

Multiplex fluorescent PCR was applied to 173 sputum specimens and compared with the results of the conventional culture method. Three bacterial-positive samples were mixed manually as positive controls. For samples with multiplex PCR results inconsistent with the culture method, targeted high-throughput sequencing (t-NGS) was performed at Hangzhou Dean Medical Laboratory Center(Details on the t-NGS procedure can be found in the S2 File). The results of bacterial culture and t-NGS were defined as the reference method, Specifically, A sample is considered positive if both culture and multiplex PCR are positive, or if there is a discrepancy between culture and multiplex PCR, and tNGS is positive. A sample is considered negative if both culture and multiplex PCR are negative, or if there is a discrepancy between culture and multiplex PCR, and tNGS is negative. All methods were carried out in accordance with relevant guidelines and regulations.

### Statistical analysis

The concordance between real-time PCR results and the results of the bacterial culture method + t-NGS was verified using Cohen's kappa test with SPSS 22.0 software. Kappa coefficient (95% CI) values were categorized as follows: 0–0.20,

slight; 0.21–0.40, fair; 0.41–0.60, moderate; 0.61–0.80, substantial; and 0.81–1.0, almost perfect agreement. A *p*-value < 0.05 was considered statistically significant [16].

## Ethics statement

This study was approved by the Ethics Committee of Jinhua Central Hospital (Approval No.: (Research) 2021-Ethics Review-254). A waiver of informed consent was granted by the Ethics Committee of Jinhua Central Hospital and the Affiliated Jinhua Hospital, Zhejiang University School of Medicine, as the study poses no ethical risk. All clinical sample analyses were conducted anonymously in accordance with the ethical review measures for biomedical research involving humans by the National Health Commission of the People's Republic of China and the Declaration of Helsinki.

## Result

### Sensitivity testing

The established multiplex fluorescence PCR method for *Streptococcus pneumoniae*, *Haemophilus influenzae*, and *Moraxella catarrhalis* detected gradient multiplicity dilutions of DNA from these strains. The lower limits of detection were 100 Copies/ml for *Streptococcus pneumoniae*, 20 Copies/ml for *Haemophilus influenzae*, and 50 Copies/ml for *Moraxella catarrhalis*. The amplification curves from the sensitivity tests are presented in Fig 1.

### Specificity testing

The specificity evaluation included both closely related species from the same genera (*Streptococcus*, *Haemophilus*, and *Moraxella*) and unrelated respiratory and non-respiratory bacterial pathogens. The multiplex fluorescence PCR assay demonstrated high specificity for *Streptococcus pneumoniae*, *Haemophilus influenzae*, and *Moraxella catarrhalis*, with no cross-reactivity observed in any of the tested strains.

### Repeatability test results

Repeatability tests for *Streptococcus pneumoniae*, *Haemophilus influenzae*, and *Moraxella catarrhalis* at various concentration gradients showed consistent amplification curves across replicates. The coefficients of variation for Ct values are listed in S3 Table.

### Clinical specimen testing

Multiplex fluorescence PCR and bacterial culture methods were used to detect 173 clinical samples. The information of the 173 participants from whom the samples were collected is shown in Table 2. In cases of discordance between the two methods, tNGS was performed. The distributions of single, double, and mixed bacterial infections detected by the three methods are shown in Table 3. Multiplex fluorescence PCR identified 15 single-positive *Streptococcus pneumoniae* cases, 37 single-positive *Haemophilus influenzae* cases, and 10 single-positive *Moraxella catarrhalis* cases. Additionally, 20 cases of double-positive *Streptococcus pneumoniae* and *Haemophilus influenzae*, 11 cases of double-positive *Moraxella catarrhalis* and *Haemophilus influenzae*, 2 cases of double-positive *Moraxella catarrhalis* and *Streptococcus pneumoniae*, and 9 cases of triple-positive samples were detected. The overall positive rates were 26% for *Streptococcus pneumoniae*, 45% for *Haemophilus influenzae*, and 18% for *Moraxella catarrhalis* are shown in Table 4. Compared to the reference method (bacterial culture + tNGS), the multiplex fluorescence PCR method demonstrated sensitivities of 97.87% for *Streptococcus pneumoniae*, 100% for *Haemophilus influenzae*, and 100% for *Moraxella catarrhalis*, with specificities of 100% for all three bacteria (Table 5). The concordance between the multiplex fluorescence PCR method and the reference method was Kappa = 0.819, *P* < 0.001, 95% CI (0.692, 0.946). The amplification plots for positive and negative clinical specimens are shown in Fig 2. Additionally, we conducted a direct comparison between multiplex PCR and culture

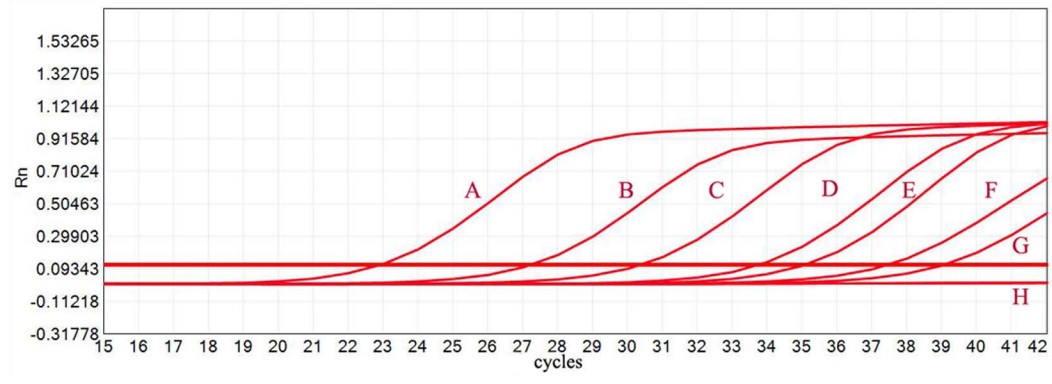

*Streptococcus pneumoniae*

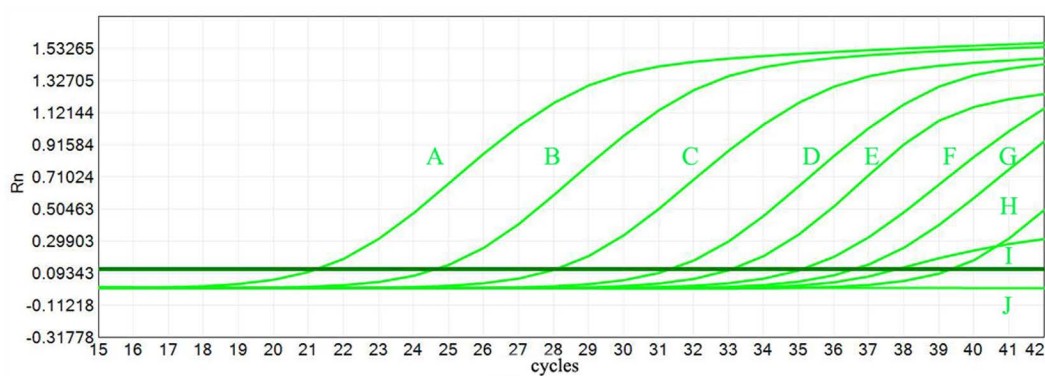

*Haemophilus influenzae*

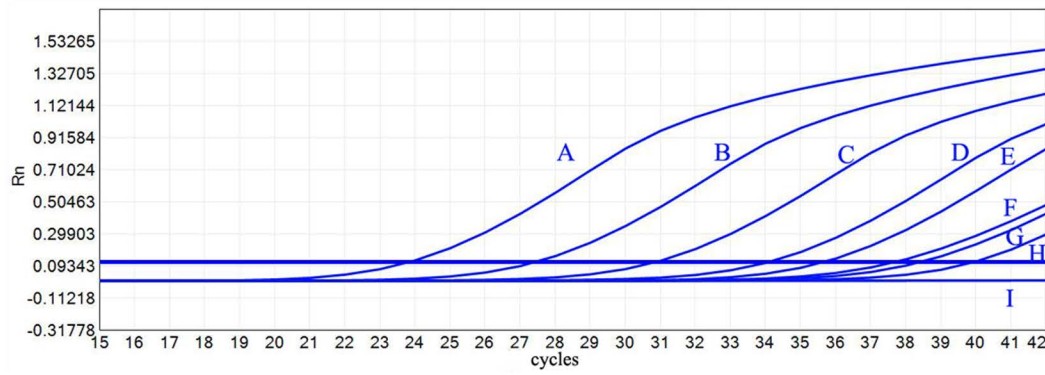

*Moraxella catarrhalis*

**Fig 1. Sensitivity analysis of the multiplex fluorescence PCR method for three fastidious bacteria.** A: 1.5 × 10^6 Copies/ml, B: 1.5 × 10^5 Copies/ml, C: 1.5 × 10^4 Copies/ml, D: 1.5 × 10^3 Copies/ml, E: 500 Copies/ml, F: 200 Copies/ml, G: 100 Copies/ml, H: 50 Copies/ml, I: 20 Copies/ml, J: 10 Copies/ml. Limit of detection for *Streptococcus pneumoniae*: 100 Copies/ml. Limit of detection for *Haemophilus influenzae*: 20 Copies/ml. Limit of detection for *Moraxella catarrhalis*: 50 Copies/ml.

**Table 2. The information of the 173 participants.**

| Category | Number of Participants | Age (years) |
|---|---|---|
| Total | 173 | 60.2±21.5 |
| Male | 121 | 59.6±22.8 |
| Female | 52 | 60.5±19.7 |
| No Antibiotics Before Sampling | 90 | 57.9±23.1 |
| Antibiotics Before Sampling | 83 | 59.6±22.9 |

**Table 3. Distribution of three pathogenic bacteria in clinical specimens detected by bacterial culture, multiplex fluorescence PCR, and tNGS methods.**

| Pathogen | Bacteriological Culture | Multiplex Fluorescent PCR | tNGS* |
|---|---|---|---|
| SP | 24 | 15 | 3 |
| HI | 52 | 37 | 7 |
| MC | 15 | 10 | – |
| HI+SP | 2 | 20 | 14 |
| MC+HI | 2 | 11 | 9 |
| MC+SP | – | 2 | 2 |
| MC+HI+SP | – | 9 | 9 |
| Negative | 78 | 69 | 1 |
| Total | 173 | 173 | 45 |

*Only samples with multiplex fluorescent PCR and culture noncompliance were tested.

tNGS: targeted high-throughput sequencing.

**Table 4. Positive rates of three pathogenic bacteria by bacterial culture, bacterial culture+tNGS, and multiplex fluorescence PCR methods in clinical specimens.**

| Methods | Total number | SP | | HI | | MC | |
|---|---|---|---|---|---|---|---|
| | | Positive (%) | Negative (%) | Positive (%) | Negative (%) | Positive (%) | Negative (%) |
| Culture | 173 | 26 (15) | 147 (85) | 56 (32) | 117 (68) | 17 (10) | 156 (90) |
| Culture+tNGS* | 173 | 47 (27) | 126 (73) | 77 (45) | 96 (55) | 32 (18) | 141 (82) |
| multiplex fluorescent PCR method | 173 | 46 (26) | 127 (74) | 77 (45) | 96 (55) | 32 (18) | 141 (82) |

*Only samples with multiple fluorescent PCR and culture noncompliance were tested.

tNGS: targeted high-throughput sequencing SP: *Streptococcus pneumoniae* HI: *Haemophilus influenzae* MC: *Moraxella catarrhalis*.

**Table 5. Comparison of multiplex fluorescence PCR results with bacterial culture+tNGS (reference method).**

| Bacterial pathogens | Multiplex fluorescent PCR method | Culture+tNGS* | | Sensitivity (%) | Specificity (%) | PPV % | NPV % |
|---|---|---|---|---|---|---|---|
| | | Positive | Negative | | | | |
| SP | Positive | 46 | 0 | 97.87 | 100 | 100 | 99.21 |
| | Negative | 1 | 126 | | | | |
| HI | Positive | 77 | 0 | 100 | 100 | 100 | 100 |
| | Negative | 0 | 96 | | | | |
| MC | Positive | 32 | 0 | 100 | 100 | 100 | 100 |
| | Negative | 0 | 141 | | | | |

*Only samples with multiplex fluorescent PCR and culture noncompliance were tested.

tNGS: targeted high-throughput sequencing SP: *Streptococcus pneumoniae* HI: *Haemophilus influenzae* MC: *Moraxella catarrhalis*.

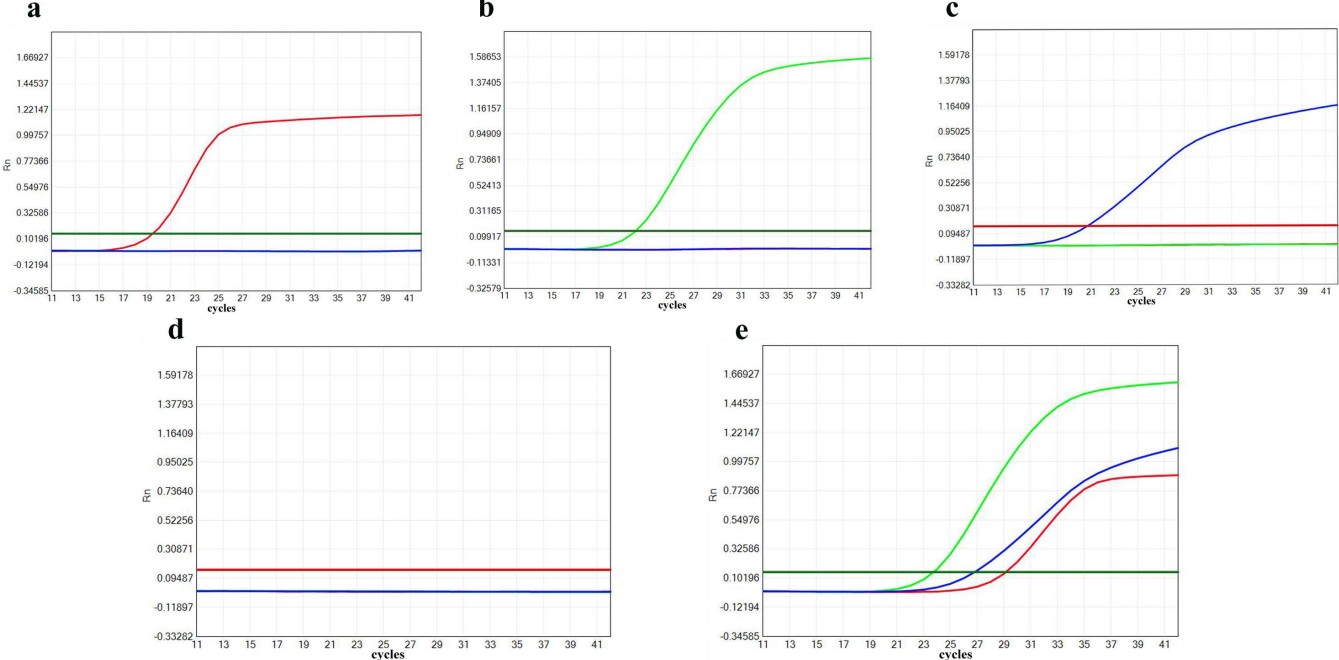

**Fig 2. Amplification plots of representative positive and negative clinical specimens using the multiplex fluorescent PCR assay.** Red, green, and blue fluorescence curves represent *Streptococcus pneumoniae* (SP), *Haemophilus influenzae* (HI), and *Moraxella catarrhalis* (MC), respectively. Panel A shows an SP-only positive sample (red curve only), Panel B an HI-only positive sample (green curve only), Panel C an MC-only positive sample (blue curve only), Panel D a negative control (no amplification), and Panel E a triple-positive sample with red, green, and blue curves.

methods to further assess the diagnostic accuracy and reliability of multiplex PCR in clinical settings. This comparison is shown in Tables 6 and 7. For *Streptococcus pneumoniae, Haemophilus influenzae,* and *Moraxella catarrhalis*, the χ² tests show a significant difference between Culture and Multiplex PCR ($P < 0.05$). In summary, our findings are illustrated in Fig 3, which provides an overview of the multiplex PCR workflow and results.

## Discussion

Respiratory infections are common in children and, while usually not fatal, can significantly impact their health [17]. Acute otitis media, a frequent complication of respiratory infections, is particularly prevalent among children and often involves various microorganisms, including *Streptococcus pneumoniae*, *Haemophilus influenzae*, and *Moraxella catarrhalis*. It is estimated that nearly 80% of young children experience at least one episode of acute otitis media each year. The over-use and unnecessary prescription of antibiotics is especially common in pediatric practice [18]. Antibiotic therapy not only targets pathogenic microorganisms but also disrupts the normal microbiome. In the upper respiratory tract, this can lead to

**Table 6. Comparative Table (Culture vs. Multiplex PCR).**

| Method | Total Number | SP Positive (%) | SP Negative (%) | HI Positive (%) | HI Negative (%) | MC Positive (%) | MC Negative (%) |
|---|---|---|---|---|---|---|---|
| Culture | 173 | 15.0 | 85.0 | 32.0 | 68.0 | 10.0 | 90.0 |
| Multiplex PCR | 173 | 26.6 | 73.4 | 44.5 | 55.5 | 18.5 | 81.5 |

SP: *Streptococcus pneumoniae* HI: *Haemophilus influenzae* MC: *Moraxella catarrhalis*.

**Table 7. χ2 Test Results for Culture vs. Multiplex PCR.**

| Bacterial Pathogen | χ2 | P-value |
|---|---|---|
| *Streptococcus pneumoniae* | 11.49 | 0.0007 |
| *Haemophilus influenzae* | 6.21 | 0.013 |
| *Moraxella catarrhalis* | 6.91 | 0.009 |

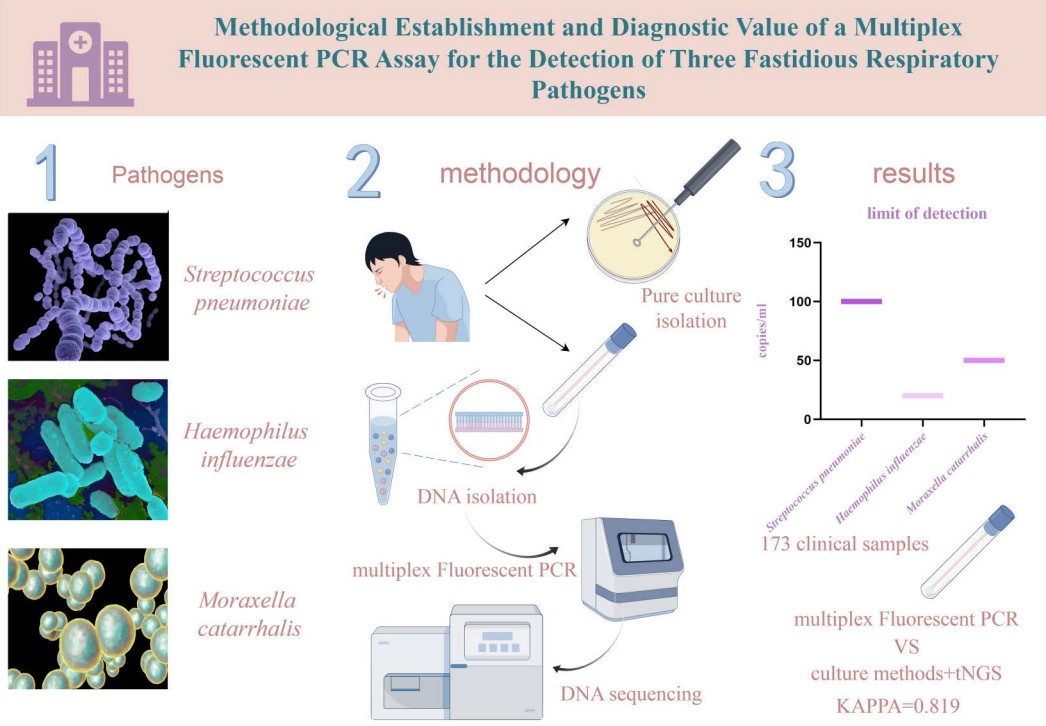

**Fig 3. Overview of Multiplex PCR Workflow and Results (by FigDraw).** This figure was created using FigDraw (www.figdraw.com). The images and elements are original and have been authorized for use. This figure was drawn by the first author, Jin Chao Shi.

an increase in pathogenic bacteria such as Haemophilus, Streptococcus, and *Moraxella catarrhalis*, while decreasing beneficial microorganisms [19]. Additionally, initiating antibiotic therapy before sample collection and analysis can reduce the likelihood of detecting positive specimens through culture methods, posing a burden on children, families, and the healthcare system. Therefore, the development of rapid and accurate diagnostic methods is essential to alleviate this burden.

Although culture methods are considered the reference method for pathogen detection, their long turnaround time and limited sensitivity reduce their utility in clinical practice. Our study focused on *Streptococcus pneumoniae*, *Haemophilus influenzae*, and *Moraxella catarrhalis*, all of which are fastidious bacteria. While routine identification of *Streptococcus pneumoniae* typically involves morphology observation and optochin sensitivity on blood agar, definitive results are not always achieved. For instance, some strains like Streptococcus pseudopneumoniae may exhibit atypical results, showing optochin susceptibility in atmospheric conditions [20,21]. *Haemophilus influenzae* requires specific growth conditions, including the presence of V and X factors, necessitating stringent culture conditions and experienced professionals. Additionally, distinguishing *Haemophilus influenzae* from other *Haemophilus* species, such as *Haemophilus haemolyticus*, can

be challenging. Interestingly, *Haemophilus haemolyticus* can inhibit the growth of *Haemophilus influenzae* by restricting its ability to acquire iron, thereby providing a protective advantage against *Haemophilus influenzae* [22,23]. This study aimed to establish a TaqMan probe multiplex fluorescent PCR assay for *Streptococcus pneumoniae*, *Haemophilus influenzae*, and *Moraxella catarrhalis* and evaluate its clinical diagnostic potential.

RT-PCR has demonstrated high sensitivity and specificity in pathogen detection, making it a promising tool for rapid screening and identification. This method targets pathogen-specific genetic material rather than antigens or antibodies, bypassing conventional culture requirements [24]. The multiplex PCR technique, an advancement of conventional PCR, allows simultaneous identification of multiple pathogens in a single reaction by mixing multiple primers and probes [25].

Previously, several studies have reported multiplex PCR methods for the detection of these pathogens [4,26,27]. Compared to these studies, our assay offers several distinct advantages. Our study employs TaqMan probes specifically designed for this research, providing higher specificity and sensitivity compared to traditional PCR methods. These probes can differentiate between closely related sequences and reduce false positives, enhancing the accuracy of pathogen detection. Our assays demonstrate excellent sensitivity and specificity, having undergone extensive specificity testing with a wide range of related and unrelated species to ensure accurate identification of the target pathogens without cross-reactivity. The sensitivity of our assays allows for the detection of low pathogen loads, making them suitable for early and accurate diagnosis. Clinically, our research targets fastidious organisms—*Streptococcus pneumoniae, Haemophilus influenzae,* and *Moraxella catarrhalis*—that are commonly encountered in clinical settings. Our method provides a reliable and rapid identification, significantly improving clinical efficiency, especially in respiratory infections where timely and accurate diagnosis is crucial for effective patient management. Additionally, we employed MGB TaqMan probes, which reduce background signal interference and stabilize probe-template hybridization, enhancing amplification efficiency. We have also demonstrated that different concentrations of primers for these three pathogens do not affect the quantitative results, ensuring robustness and reliability in various clinical scenarios. These advantages underscore the practical value of our assay in clinical diagnostics. Notably, the ability to detect low bacterial DNA copy numbers (e.g., < 100 copies/mL) is critical for identifying early-stage infections, monitoring patient post-antibiotic treatment, and detecting pathogens in polymicrobial or low-burden infections where culture often fails. While the high sensitivity raises the question of potential over-detection, the inclusion of strict negative controls and confirmation by t-NGS minimizes the risk of false positives, ensuring diagnostic accuracy. Thus, our method provides a robust and clinically meaningful improvement over traditional approaches.

Comparing traditional culture methods and tNGS technology, we designed the multiplex fluorescence PCR method to accurately identify target bacterial DNAs without cross-reactivity, enabling simultaneous and rapid detection of *Streptococcus pneumoniae*, *Haemophilus influenzae*, and *Moraxella catarrhalis*. Sensitivity tests showed detection limits of 100 Copies/ml for *Streptococcus pneumoniae*, 20 Copies/ml for *Haemophilus influenzae*, and 50 Copies/ml for *Moraxella catarrhalis*.

We tested 173 clinical samples to assess the method's clinical utility. The multiplex fluorescent PCR method detected mixed infections that conventional culture methods missed, possibly due to overlooked or non-growing single colonies. This supports previous findings of multiple pathogen infections in hospitalized patients [28]. The high frequency of co-occurrence of *Streptococcus pneumoniae*, *Haemophilus influenzae*, and *Moraxella catarrhalis* in respiratory samples is likely due to their common habitat and similar modes of transmission. These pathogens often colonize the nasopharynx and upper respiratory tract, particularly in children and the elderly. They can exist as part of the normal flora but can also become pathogenic under certain conditions, such as a weakened immune system or viral co-infections. This co-colonization can lead to polymicrobial infections, where the presence of one pathogen may facilitate the colonization or persistence of another. Additionally, these bacteria share similar risk factors and modes of transmission, such as respiratory droplets, contributing to their frequent co-occurrence [29]. Including tNGS in our study, we found that this technology, covering a wide range of pathogens, is valuable for identifying rare and hard-to-diagnose infections [30]. The multiplex

fluorescent PCR method achieved the highest positive detection rate for *Haemophilus influenzae* (45%). *Streptococcus pneumoniae* and *Haemophilus influenzae* were most commonly detected in mixed infections. The Kappa value of 0.819 indicated high agreement between the PCR method and the reference method. Additionally, we conducted a direct comparison between multiplex PCR and culture methods to further assess the diagnostic accuracy and reliability of multiplex PCR, These results suggest a statistically significant difference in the detection rates of these pathogens when comparing Culture and Multiplex PCR methods, highlighting the potential of the multiplex PCR method to complement or replace traditional culture methods in clinical diagnostics.

In clinical specimen testing, the multiplex real-time fluorescent PCR assay provides a practical means for the rapid detection and identification of infections caused by *Streptococcus pneumoniae*, *Haemophilus influenzae*, and *Moraxella catarrhalis*. This method, when combined with bacterial culture and drug sensitivity tests, enhances diagnostic accuracy, reduces antibiotic misuse, and supports clinical precision medicine. However, our study has several limitations. First, while qPCR is inherently a quantitative method, in our study, it was used primarily for qualitative detection. We did not include the necessary controls, such as 16S rRNA real-time PCR, to measure the relative abundance of different target sequences. This limits the ability to compare their relative importance in the samples. However, qualitative detection is generally more cost-effective and faster, making it more suitable for rapid clinical diagnostics. Clinicians must consider the sample source and patient condition to distinguish between colonization and infection. Second, a strain of *Streptococcus pneumoniae* in our clinical sample tested positive by culture but negative by our multiplex PCR method. This discrepancy might be due to a low concentration of the target strain, below the PCR assay's sensitivity threshold, or a sequence variant in the target gene, leading to unstable primer binding or reduced amplification efficiency. Additionally, the presence of bile-insoluble S. pneumoniae strains, which are an infrequent pneumococcal phenotype, can lead to false-negative results in PCR assays targeting the lytA gene, as these strains do not match the primers and probes used in the assay. Bile solubility test is a specific test for the function of autolysin, which is a product of the lytA gene transcription. Changes in the lytA gene sequence may affect the results of the bile solubility test, thereby impacting the PCR results [31].Third, the use of representative strains in this study may not fully capture the diversity and characteristics of the related species as a whole [32]. We plan to optimize our PCR method to enhance detection sensitivity for low-concentration samples and improve primer design to address potential sequence variants. We will also strengthen sample processing quality control to ensure purity and integrity. Future work will focus on refining these quantitative measures to better distinguish between potential colonization and true infection. We also plan to incorporate additional clinical data and correlate PCR findings with patient outcomes to enhance the accuracy of our diagnostic approach. Through these efforts, we aim to improve the accuracy and reliability of our methods, providing more dependable support for clinical diagnosis.

## Conclusion

This study validated the TaqMan probe multiplex fluorescent PCR technique for rapid detection of *Streptococcus pneumoniae*, *Haemophilus influenzae*, and *Moraxella catarrhalis*. By comparing it with traditional culture and tNGS methods, we demonstrated the method's efficiency and accuracy, especially in detecting mixed infections. This tool offers clinicians a reliable and rapid pathogen detection method, reducing diagnostic costs and positively impacting children's health and clinical care quality.

## Supporting information

**S1 Table. Main reagents and instruments used in the study.** This table lists the PCR reagents, primer suppliers, DNA extraction kits, and PCR instruments.
(DOCX)

**S2 File. tNGS workflow on the Nanopore platform. This file describes the experimental procedure for targeted nanopore sequencing used to verify multiplex PCR discordant results.**
(DOCX)

**S3 Table. Repeatability analysis of TaqMan probe-based multiplex fluorescent PCR at different template concentrations.** This includes Ct values, means, standard deviations, and coefficients of variation (CV) for each pathogen.
(DOCX)

## Author contributions

**Conceptualization:** Jingchao Shi.

**Data curation:** Jingchao Shi, Shuyun Chen.

**Funding acquisition:** Yijun Zhu, Xiaoyun Shan.

**Investigation:** Yijun Zhu.

**Methodology:** Jingchao Shi, Yijun Zhu, Xiaoyun Shan, Lihong Bo.

**Project administration:** Yijun Zhu.

**Resources:** Xiaoyun Shan, Lihong Bo, Keqiang Chen.

**Software:** Jingchao Shi, Shuyun Chen.

**Validation:** Jingchao Shi, Shuyun Chen, Kai Shen.

**Visualization:** Jingchao Shi, Shuyun Chen, Kai Shen, Keqiang Chen.

**Writing – original draft:** Jingchao Shi.

**Writing – review & editing:** Jingchao Shi.

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
