## [Decision Letter · Decision Letter 0]

Dear Dr. Zhu,

Thank you for submitting your manuscript to PLOS ONE. After careful consideration, we feel that it has merit but does not fully meet PLOS ONE’s publication criteria as it currently stands. Therefore, we invite you to submit a revised version of the manuscript that addresses all the points raised during the review process.

Please submit your revised manuscript by Aug 02 2025 11:59PM. If you will need more time than this to complete your revisions, please reply to this message or contact the journal office at plosone@plos.org . A rebuttal letter that responds to each point raised by the academic editor and reviewer(s). You should upload this letter as a separate file labeled 'Response to Reviewers'.A marked-up copy of your manuscript that highlights changes made to the original version. You should upload this as a separate file labeled 'Revised Manuscript with Track Changes'.An unmarked version of your revised paper without tracked changes. You should upload this as a separate file labeled 'Manuscript'.

We look forward to receiving your revised manuscript.

Kind regards,

Daniela Flavia Hozbor

Academic Editor

PLOS ONE

Journal Requirements:

“Science Technology Department of Zhejiang province, China (LGC22H200018), Jinhua Science and Technology Bureau, Zhejiang province, China (2021-3-026).”

3. Please note that funding information should not appear in the Acknowledgments section or other areas of your manuscript. We will only publish funding information present in the Funding Statement section of the online submission form. Please remove any funding-related text from the manuscript. 

Reviewers' comments:

Reviewer's Responses to Questions

**Comments to the Author**

1. Is the manuscript technically sound, and do the data support the conclusions?

Reviewer #1: Yes

Reviewer #2: Yes

2. Has the statistical analysis been performed appropriately and rigorously?

Reviewer #1: Yes

Reviewer #2: Yes

3. Have the authors made all data underlying the findings in their manuscript fully available?

Reviewer #1: Yes

Reviewer #2: Yes

4. Is the manuscript presented in an intelligible fashion and written in standard English?

Reviewer #1: Yes

Reviewer #2: Yes

Reviewer #1: Dear Authors,

I commend you for your work. Here are some comments and suggestions to further strengthen your manuscript:

Abstract

To enhance the abstract, consider adding a brief background to provide context for the study. Mention that Streptococcus pneumoniae, Haemophilus influenzae, and Moraxella catarrhalis are common fastidious bacteria causing respiratory tract infections. Highlight the challenges in their detection using traditional methods.

Introduction

1. To strengthen the introduction, consider expanding on the clinical significance of mixed infections and the challenges they pose for diagnosis and treatment

2. You might also want to include a brief discussion of the mechanisms by which these pathogens cause disease and the host immune response to these infections

Methods

1. Consider providing more information on the quality control measures taken to ensure the accuracy and reliability of the DNA extraction and PCR amplification steps

Overall, this is a well-conducted study with significant clinical implications. Addressing the comments and suggestions will further enhance the quality and impact of your manuscript.

Reviewer #2: This paper outlines a novel multiplex PCR assay for the identification of fastidious bacteria related to paediatric inner ear infections. This multiplex PCR method is shown to be sensitive down to 20-100 copies/ml within a reasonable number of PCR cycles to avoid overamplifications. This assay is shown to be specific for the three bacteria of interest compared to both species of the same genus and other unrelated bacteria. Finally, the authors show, using 173 clinical samples, that this multiplex assay outperforms traditional culture methods in the detection of concurrent bacteria within a sample, confirmed through NGS methods when in doubt. Overall, this paper lays out a novel diagnostic approach which would aid in early identification of infection causing bacteria and inform on antibiotic usage if implemented in the clinic.

Minor Revisions

1. Although the complete list of bacteria tested for specificity is given in the methodology, I suggest, as a minimum, highlighting in the results text that the bacterial strains tested included both other Streptococcus, Haemophilus and Moraxella strains as well as other unrelated infectious bacteria types. The exhaustive list is not required in the results.

2. From the text and methods it is not exactly clear what concentrations were used for the repeatability test. It would be good to have this test put in context of the sensitivity test. For example, for the concentrations used for the repeatability test, are these similar to 10 copies/ml or 500 copies/ml. This is important to understand the reproducibility of the assay at low bacterial numbers.

3. For figure 2: It is not clear, are these curves representative of 4 samples: 3 single positive and a negative? Or is it a single triple positive sample and a negative. Ideally the former would be shown alongside a triple positive for comparison, as individual graph panels.

4. From line 333: This paragraph is repetitive of the results. If leaving this section in, expand on why the ability to detect low copy numbers is important and possible discuss whether this detectability is overly sensitive.

**Do you want your identity to be public for this peer review?** For information about this choice, including consent withdrawal, please see our Privacy Policy

Reviewer #1: No

Reviewer #2: No

---

## [Author Response · Author response to Decision Letter 1]

25 Jun 2025

Journal: PLOS ONE 

Manuscript title: Methodological Establishment and Diagnostic Value of a Multiplex Fluorescent PCR Assay for the Detection of Three Fastidious Respiratory Pathogens

Manuscript ID: PONE-D-24-54222

Authors: Jingchao Shi, Yijun Zhu, Shuyun Chen, Xiaoyun Shan, Lihong Bo, Kai Shen, Keqiang Chen

Dear editor and reviewers

We appreciate your assessments of our manuscript (Manuscript ID: PONE-D-24-54222), entitled “Methodological Establishment and Diagnostic Value of a Multiplex Fluorescent PCR Assay for the Detection of Three Fastidious Respiratory Pathogens”, and have found that the reviewers’ comments and suggestions are valuable in revising the manuscript. We have made the necessary revisions as recommended by the reviewers. In the revised manuscript, we have highlighted all the amendments using yellow text.

Thank you once again, and please don't hesitate to contact us if you have any further comments or questions. We are looking forward to your final decision.

Thank you very much for your great help!

Sincerely,

Jingchao Shi

The following are our point-by-point responses, in order of the comments about the manuscript:

Reviewer #1:

1. Abstract

To enhance the abstract, consider adding a brief background to provide context for the study. Mention that Streptococcus pneumoniae, Haemophilus influenzae, and Moraxella catarrhalis are common fastidious bacteria causing respiratory tract infections. Highlight the challenges in their detection using traditional methods.

Introduction

Response We thank the reviewer for this constructive suggestion. In response, we have revised the abstract to include a brief background that highlights the clinical importance of Streptococcus pneumoniae, Haemophilus influenzae, and Moraxella catarrhalis as fastidious pathogens commonly associated with respiratory tract infections, and the limitations of traditional culture-based diagnostic methods. (line15-19)

2. Introduction

To strengthen the introduction, consider expanding on the clinical significance of mixed infections and the challenges they pose for diagnosis and treatment.

You might also want to include a brief discussion of the mechanisms by which these pathogens cause disease and the host immune response to these infections

Response We appreciate the reviewer’s valuable comments. In response, we have revised the Introduction section to provide a more comprehensive background. We have now added discussion on (1) the clinical significance of mixed infections and their diagnostic and therapeutic challenges, and (2) the pathogenic mechanisms and host immune responses associated with S. pneumoniae, H. influenzae, and M. catarrhalis. These changes enhance the rationale for developing a rapid and accurate detection method for these pathogens. (line61-76)

3. Methods

Consider providing more information on the quality control measures taken to ensure the accuracy and reliability of the DNA extraction and PCR amplification steps

Overall, this is a well-conducted study with significant clinical implications. Addressing the comments and suggestions will further enhance the quality and impact of your manuscript.

Response We appreciate the reviewer’s insightful comment regarding the need to elaborate on quality control measures in the Materials and Methods section. To address this, we have added specific details on the quality assurance steps taken during DNA extraction, PCR setup, and amplification, including the use of internal controls, reagent handling protocols, and contamination prevention practices. These additions enhance the transparency and reproducibility of our methods and underscore the reliability of our assay. (line113-116, line133-134,line152-158, line 205-206)

Reviewer #2:

1. Although the complete list of bacteria tested for specificity is given in the methodology, I suggest, as a minimum, highlighting in the results text that the bacterial strains tested included both other Streptococcus, Haemophilus and Moraxella strains as well as other unrelated infectious bacteria types. The exhaustive list is not required in the results.

Response We thank the reviewer for the suggestion. We have now added a summary in the results section indicating that the specificity panel included closely related species from the same genera as the targets (e.g., Streptococcus mitis, Haemophilus haemolyticus, Moraxella osloensis), as well as other common respiratory and unrelated bacterial pathogens. (line245-249)

The full list of bacterial species used for specificity testing was provided in the Methods section to ensure methodological transparency and reproducibility. This allows other researchers to replicate the study or assess potential cross-reactivity with specific organisms of interest.

2. From the text and methods it is not exactly clear what concentrations were used for the repeatability test. It would be good to have this test put in context of the sensitivity test. For example, for the concentrations used for the repeatability test, are these similar to 10 copies/ml or 500 copies/ml. This is important to understand the reproducibility of the assay at low bacterial numbers.

Response Thank you for your insightful comment. We have clarified the relationship between the concentrations used in the repeatability test and those in the sensitivity analysis.

In the repeatability test, three concentration levels were selected for each pathogen, representing high, moderate, and low template amounts. These corresponded approximately to Ct values of ~24, ~28–29, and ~31–35, respectively. Based on the standard curve generated during the sensitivity test, these Ct values are estimated to correspond to the following DNA concentrations:

Ct ≈ 24: ~10⁵–10⁶ copies/mL

Ct ≈ 28–29: ~10³–10⁴ copies/mL

Ct ≈ 31–35: ~10–100 copies/mL

Therefore, the repeatability test included low concentrations close to the detection limit (10–20 copies/mL), ensuring assessment of assay reproducibility across the entire analytical range. The detailed Ct values and corresponding coefficients of variation (CVs) are provided in Supplementary Table 3, showing low intra-assay variability (CVs from 0.03 to 0.21), even at low template levels.

We have also added this explanation to the Methods section for clarity. (line198-206)

3. For figure 2: It is not clear, are these curves representative of 4 samples: 3 single positive and a negative? Or is it a single triple positive sample and a negative. Ideally the former would be shown alongside a triple positive for comparison, as individual graph panels.

Response We appreciate the reviewer’s helpful suggestion regarding the clarification of Figure 2. In the revised version of Figure 2, we have included individual amplification curves for five representative samples:

Panel A: S. pneumoniae-only positive

Panel B: H. influenzae-only positive

Panel C: M. catarrhalis-only positive

Panel D: Negative control

Panel E: Triple-positive sample (simultaneous presence of all three pathogens)

Each panel clearly shows the fluorescence amplification signal for the respective pathogen(s), and the inclusion of both single and mixed infections enhances the clarity and interpretability of the assay’s multiplex capability.

We believe this revision more effectively illustrates the specificity and multiplex detection capacity of our assay and fully addresses the reviewer’s concern.

4. From line 333: This paragraph is repetitive of the results. If leaving this section in, expand on why the ability to detect low copy numbers is important and possible discuss whether this detectability is overly sensitive.

Response We appreciate the reviewer’s insightful comment. We agree that the original paragraph in line 333 was somewhat repetitive and did not adequately explain the clinical relevance of detecting low bacterial copy numbers. In response, we have revised this section of the manuscript to highlight why this feature is important in clinical practice. Specifically, we emphasize that the ability to detect low-copy-number pathogens can aid in the diagnosis of early-stage infections, cases with low bacterial loads due to prior antibiotic treatment, or polymicrobial infections where certain pathogens are present in minor proportions.

We have also addressed the concern of potential over-sensitivity. The inclusion of appropriate negative controls and the use of t-NGS confirmation in discordant cases help minimize the risk of false positives due to contamination or nonspecific amplification. Therefore, we believe that the assay maintains a clinically relevant balance between high sensitivity and specificity. (line371-379)

---

## [Decision Letter · Decision Letter 1]

Methodological Establishment and Diagnostic Value of a Multiplex Fluorescent PCR Assay for the Detection of Three Fastidious Respiratory Pathogens

PONE-D-24-54222R1

Dear Dr. Yijun Zhu,

We’re pleased to inform you that your manuscript has been judged scientifically suitable for publication and will be formally accepted for publication once it meets all outstanding technical requirements.

Kind regards,

Daniela Flavia Hozbor

Academic Editor

PLOS ONE

Additional Editor Comments (optional):

Reviewers' comments:

Reviewer's Responses to Questions

**Comments to the Author**

Reviewer #1: All comments have been addressed

Reviewer #2: All comments have been addressed

2. Is the manuscript technically sound, and do the data support the conclusions?

Reviewer #1: Yes

Reviewer #2: Yes

3. Has the statistical analysis been performed appropriately and rigorously?

Reviewer #1: N/A

Reviewer #2: Yes

4. Have the authors made all data underlying the findings in their manuscript fully available?

Reviewer #1: Yes

Reviewer #2: Yes

5. Is the manuscript presented in an intelligible fashion and written in standard English?

Reviewer #1: Yes

Reviewer #2: Yes

Reviewer #1: Thank you for addressing the comments, I think your paper is now suitable for publication. Congratulations.

Reviewer #2: (No Response)

**Do you want your identity to be public for this peer review?** For information about this choice, including consent withdrawal, please see our Privacy Policy

Reviewer #1: No

Reviewer #2: No

---

## [Editor Report · Acceptance letter]

PONE-D-24-54222R1

PLOS ONE

Dear Dr. Zhu,

I'm pleased to inform you that your manuscript has been deemed suitable for publication in PLOS ONE. Congratulations! Your manuscript is now being handed over to our production team.

Kind regards,

on behalf of

Dr. Daniela Flavia Hozbor

Academic Editor

PLOS ONE